# Self-Organized Group for Cooperative Multi-agent Reinforcement Learning

**Jianzhun Shao, Zhiqiang Lou, Hongchang Zhang, Yuhang Jiang, Shuncheng He,
Xiangyang Ji**
Department of Automation
Tsinghua University, Beijing, China
{sjz18, lzq20, hc-zhang19, jiangyh19, hesc16}@mails.tsinghua.edu.cn
xyji@tsinghua.edu.cn

## Abstract

Centralized training with decentralized execution (CTDE) has achieved great success in cooperative multi-agent reinforcement learning (MARL) in practical applications. However, CTDE-based methods typically suffer from poor zero-shot generalization ability with dynamic team composition and varying partial observability. To tackle these issues, we propose a spontaneously grouping mechanism, termed Self-Organized Group (SOG), which is featured with a conductor election (CE) and a message summary (MS) mechanism. In CE, a certain number of conductors are elected every $T$ time-steps to temporally construct groups, each with conductor-follower consensus where the followers are constrained to only communicate with their conductor. In MS, each conductor summarize and distribute the received messages to all affiliate group members to hold a unified scheduling. SOG provides zero-shot generalization ability to the dynamic number of agents and the varying partial observability. Sufficient experiments on mainstream multi-agent benchmarks exhibit superiority of SOG.

## 1 Introduction

Cooperative multi-agent deep reinforcement learning algorithms (MARL) have been ubiquitously applied to real-world scenarios, such as autonomous vehicle teams [45], sensor networks [48], and social science [13]. Recently MARL has gained extraordinary performance in various multiplayer games like Dota [3], StartCraft [30] and soccer [16]. The framework of centralized training with decentralized execution (CTDE) [8, 28] is one of the popular frameworks for solving cooperative multi-agent tasks. Centralized training renders the CTDE paradigm better agent cooperation while independent execution endows the multi-agent system with more efficiency and scalability.

Classical CTDE algorithms like QMIX [28] and MADDPG [21] are confined to the fixed size of agents. However, the number of involved agents tends to vary in real-world multi-agent scenarios. Recently, some methods introduce the attention mechanism [40] to train on the varying number of agents simultaneously [12, 11]. They only seek for a solution covering a range of varying team size, unable to provide the generalization ability to the cases not in the range. Agarwal et al. [1] and Liu et al. [18] further introduce the communication mechanism to provide the adaptability on the dynamic team composition (i.e., the team size varies).

In the paper, we propose a method called "Self-Organized Group (SOG)"to possess zero-shot generalization for multi-agent reinforcement learning, which provides strong adaptability to the scenarios with varying number of agents and even varying sight of agents. In spirit of the individuals' cooperation in social science [10] and the leader-following formation in multi-agent system [9], part of the agents are elected as conductors at intervals, and for each conductor a temporary group is

36th Conference on Neural Information Processing Systems (NeurIPS 2022).

formed with conductor-follower consensus. In each group, the followers can only communicate with the conductor, while the conductor summarizes all the received messages and distributes the refined message to all group members aiming for a unified target. Our insight is that, an organized group under the unified command of a conductor can better adapt to an unseen scenario than individuals. To be specific, each agent's behavior is prone to diverging from the training pattern when presented with an unseen scenario. They may behave in contradictory ways and pose a threat to the system's stability. On the contrary, a conductor could send unified commands, making each agent behave in order, thus avoiding the overall collapse of the training pattern. In the meantime, our method largely reduces the bandwidth and the cost required for communication. For efficient and economical message passing, we also design a variational message summarizer (MS), which compresses the local observation information and helps predict the future trajectories.

Since the conductor navigates the information propagation in the whole group, Conductor Election (CE) plays a critical role in SOG. Except for the simple random CE which satisfies the fully decentralized execution paradigm, we further propose a heuristic variant: determinantal point process (DPP) [22]-based CE, and a learning-based variant: policy gradient (PG) based CE. DPP-based CE chooses conductors in the principle of maximum entropy and avoids the occurrence of homogeneous conductors. PG-based CE formulates the CE as a learning process and takes the long-term benefits of the selection into consideration.

We conduct experiments on three commonly used multi-agent benchmarks, including a resource collection task, a predator-prey task, and a set of customized StarCraft micromanagement tasks. To validate the zero-shot generalization ability of our method, we evaluate trained models in more complicated scenarios than training. We not only increase the number of agents for evaluation, but also adjust the agents' utility, e.g., reducing the sight range of each agent. The results show that SOG has better zero-shot generalization ability, not only for the dynamic team composition, but also for the unseen environment condition, than current state-of-the-art methods on all three benchmarks.

The contribution of our paper is summarized as follows: (1) We propose Self-Organized Group, a mechanism for enhancing agents' zero-shot generalization ability. (2) We design two conductor election ways for reasonable group forming and a variational message summarizer for efficient and economical message passing. (3) We achieve better generalization ability on three commonly used multi-agent benchmarks than current state-of-the-art methods.

## 2   Related Work

**Centralized Training with Decentralized Execution**. We mainly focus on the typical paradigm of centralized training with decentralized execution (CTDE) for cooperative multi-agent tasks in this part. It combines the advantages of independent Q-learning [38] and joint action learning [4]. A series of work concentrate on factorizing Q functions [35, 28] and deriving theoretical guarantee for the policy optimality [33, 41, 29]. Some other works use actor-critic methods with a centralized critic [21, 7, 50]. Although the CTDE paradigm shows great empirical performance in many multi-agent tasks, it may fail on some simple situations without communication on execution due to the partial observability of the local agent [18]. In addition, traditional CTDE methods usually concentrate on invariable environment conditions and fixed team size.

**Communication for MARL**. Many efforts have been made for communicating agents in multi-agent systems [32]. Recently some works assume the information can propagate among homogeneous agents during decentralized execution [6, 36, 49, 34, 19], some of which use the graph neural network to handle the message passing [23, 24, 26, 24]. Some works focus on communication between heterogenous agents [25, 31], but the role of each agent is usually fixed, therefore limiting the model's generalization ability. These works usually pre-define the number of agents, and can hardly handle the dynamic team composition. Liu et al. [18] uses a predefined coach with global information to instruct dynamic number of agents. However, a global coach that can communicate with all agents is not often available in real-world domains. In contrast to these methods, our method enables all agents to become a conductor, while reserves the efficiency and scalability of CTDE.

**Transfer Learning and Curriculum Learning**. An important topic for multi-agent systems is the adaptation to the mutable environment. A bunch of works attempt to deal with the increasingly challenging environment by designing a set of curricula [2, 20, 44]. Another line of works utilize the attention mechanism or graph neural networks to handle the diversity of the local observation [14,

1, 11, 12]. Wang et al. [43] allocates roles with limited action space to agents, and the study shows the model's ability of task transfer by increasing the action space for role clusters. Iqbal et al. [12] trains simultaneously on multiple tasks by randomly masking out some entities the agent observes. Unlike these methods, our method bypasses the overlarge training burden of curriculum learning, and is more robust in the dynamic team composition and varying environment conditions.

## 3 Background

In our work, we consider a fully collaborative multi-agent task with $n$ agents, which can be modeled as a *decentralised partially observable Markov decision process* (Dec-POMDP) [27] $G = \langle S, A, I, P, r, Z, O, n, \gamma \rangle$, where $s \in S$ is the true state of the environment. At each time step $t$, each agent $i \in I \equiv \{1, ..., n\}$ chooses an action $a_i \in A$, which is an element of a joint action $\mathbf{a} \in \mathbf{A} \equiv A^n$. $P(s_{t+1}|s_t, \mathbf{a}_t) : S \times \mathbf{A} \times S \rightarrow [0, 1]$ is the state transition function of the environment. All agents share the same reward function $r(s, \mathbf{a}) : S \times \mathbf{A} \rightarrow \mathbb{R}$. $\gamma \in [0, 1)$ is the discount factor. Due to the *partial observability*, each agent $i$ has its local observations $o^i \in O$ drawn from the observation function $Z(s, i) : S \times I \rightarrow O$. In our setting, we assume agents can communicate within its sight range. Each agent $i$ may receive a message $\mu^i \in U$ from its neighbors. Each agent chooses an action by its stochastic policy $\pi^i(a^i|\rho^i, \mu^i) : \Gamma \times U \rightarrow \Delta([0, 1]^{|A|})$, where $\rho^i \in \Gamma \equiv (O \times A)^*$ denotes the action-observation history of agent $i$, and $\boldsymbol{\rho}$ is the action-observation histories of all agents. The agents' joint policy $\pi$ induces a joint *action-value function*: $Q^\pi(s_t, \mathbf{a}_t) = \mathbb{E}_{s_{t+1:\infty}, \mathbf{a}_{t+1:\infty}}[R_t|s_t, \mathbf{a}_t]$, where $R_t = \sum_{k=0}^{\infty} \gamma^k r_{t+k}$ is the discounted accumulated reward. The goal of our method is to find the optimal joint policy $\pi^*$ such that $Q^{\pi^*}(s, \mathbf{a}) \geq Q^\pi(s, \mathbf{a})$, for all $\pi$ and $(s, \mathbf{a}) \in S \times \mathbf{A}$. In practice, to handle the input with dynamic team size, we use the entity-wise input rather than the vector input, detailed in Appendix A.

We use the popular centralized training with decentralized execution (CTDE) paradigm [8, 28] as the baseline, also allowing the communication between adjacent agents. During training, the method has the access to the full state and each agent's action-observation history. During testing (execution), each agent only has the access to its own action-observation history and communication messages from neighbors.

## 4 Method

In this section, we first highlight the importance of the communication during decentralized execution, and propose self-organized group for decentralized communication. Next we introduce the learning objective of the communication message. Finally we present the whole training procedure together with the model architecture.

### 4.1 Downsides of CTDE

Consider a simple scenario in Fig. 1(a). The task is fulfilled only when the three buttons are pressed synchronously. Therefore, agents A, B, and C need to press the three nonrecurring buttons respectively. The button 1 and 2 are ten meters apart, while button 3 is at the midpoint of them. The sight range of each agent is a circle with radius six meters. As a result, the agent who is in charge of the button 2 can't perceive if some agent is pressing the button 1, due to sight range limitation. If we use the common centralized training with de-

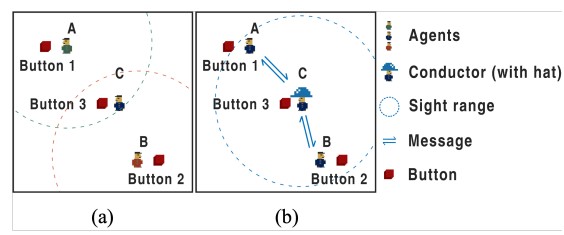

(a)  (b)

Figure 1: The illustration of a simple multi-agent scenario.

centralized execution (CTDE) method, like QMIX [28], to deal with this task, these agents can not efficiently coordinate beyond the individual sight range, since each agent's execution is conditioned on its own action-observation history. The communication between agents is one of the easy and flexible solutions to handle the partial observability.

To extend the CTDE paradigm based on the communication mechanism, we now present the Self-Organized Group (SOG), a mechanism to group all agents and select a conductor for each group.

## 4.2   Self-Organized Group

Aiming for a cooperative multi-agent task with dynamic team composition, every $T$ time-steps, we elect a certain number of conductors and the groups are constructed accordingly. In this section we first show the formulation of SOG. Then we present the message summarizer in Sec. 4.2.1, and we discuss in Sec. 4.2.2 the strategies for conductor election. An example group is illustrated as in Fig. 1(b), where agent C is appointed as the conductor, with the collected information from A and B to broaden the sight range. Since our main task lies in MARL, we simplify the communication procedure, assuming that all the messages can be delivered accurately without delay and each agent can handle all the received messages at the same time. Our goal is to build a communication mechanism that satisfies the following properties:

- **Lightweight**. The communication message should be brief and informative, e.g., a three-dimensional summary vector of the agent's current state.
- **Robust**. The group should be able to handle the varying number of group members and the unfamiliar environmental conditions (e.g. the varying sight range).

The process of group organization is as follows (Fig. 2): every $T$ time-steps, some agents are elected to be conductors. We allow multiple conductors in the same time. Then conductors send group invitations to all agents within their sight ranges. The non-conductor agents who receive one invitation then send their personal messages $\zeta$ back to the conductor. Here the personal message $\zeta$ is delicately designed by a message summarizer, described in Sec. 4.2.1. For non-conductor agents who receive multiple group invitations, they randomly choose one as the conductor and reply. The conductor and its interactive agents temporarily form a group. For those who receive no group invitation or leave the sight of the constructor during in-group communication, they form a 1-agent group of themselves.

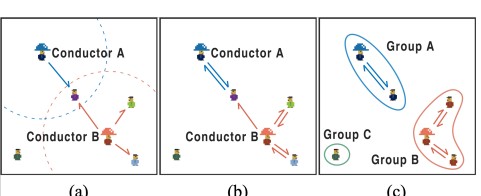

(a)       (b)       (c)

Figure 2: The process of group organization. (a) Conductors are elected, and send group invitations to the neighbors. (b) Followers choose the conductor and send the summarized messages. (c) Groups are formed and conductors send unified targets to the group members.

After the group is formed, the conductor processes the messages from all the followers (including itself). Then it sends the processed message $\xi^j$ back to each member that belongs to group $j$. Each agent only needs to communicate with its group's conductor, therefore it greatly reduces the bandwidth required for communication. We use a non-parametric message mixer for each conductor:

$$\xi^j = \frac{1}{M^j} \sum_{i=1}^{M^j} \zeta^i, i \in \text{Group}(j), \tag{1}$$

where $M^j$ is the size of group $j$. In this way, the conductor takes no extra computational cost for message mixing and distributing procedure, and each agent can play the role of the conductor without changing its network structure.

The communication takes place every $T$ time-steps. Each agent $i$ uses its local observation $o_i$ and the message $\mu^i$ to predict its action. $\mu^i$ is defined as:

$$\mu_t^i = \mathbb{I}(t, T) \cdot \xi_t^j + (1 - \mathbb{I}(t, T)) \cdot \zeta_t^i, \quad i \in \text{Group}(j), \tag{2}$$

where $\mathbb{I}(t, T)$ is the indicator whether $t$ is divisible by $T$.

Through such mechanism, agents can share the information of the whole group with the minimal communication cost and coordinate more efficiently. We present the quantitative analysis on the communication cost in Appendix H, which is $1/M$ of the cost of the fully-connected communication like Agarwal et al. [1], where $M$ is the average number of neighbors for each agent.

In unfamiliar scenarios, the conductor integrates the followers' information and all group members execute a unified command, driven by a delicate message summarizer. Such a in-group communication

mechanism empowers the system better stability compared to the non-conductor-following formation. With time-varying conductor election, SOG shows strong zero-shot generalization ability to the dynamic team composition and unseen environmental conditions.

### 4.2.1 Message Summary (MS)

After the communication protocol is built, we now present the content of the communication message. Inspired by recent works on role learning and state summary by variational inference [42, 18, 17], we propose a message summarizer for agents to summarize their local observations as brief latent variables. A summarizer can extract valuable information from the whole trajectory, and reduce the computational complexity.

We first aim to distill the information about the agent $i$' s transition of future $T$-steps. Let $\tau_t^i = (o_{t+1}^i, a_{t+1}^i, ..., o_{t+T-1}^i, a_{t+T-1}^i)$. The personal message is a random Gaussian variable of $C$ dimensions sampled from an encoder, i.e., $\zeta_t^i \sim f_\psi(o_t^i)$. We maximize the mutual information between $\zeta_t^i$ and $\tau_t^i$ conditioned on $o_t^i$. $I(\zeta_t^i; \tau_t^i | o_t^i)$ has the following lower bound:

$$I(\zeta_t^i; \tau_t^i | o_t^i) \geq \mathbb{E}_{o_t^i, \zeta_t^i, \tau_t^i} \left[ \log q_\phi(\zeta_t^i | o_t^i, \tau_t^i) \right] + H(\zeta_t^i | o_t^i), \tag{3}$$

where $q_\phi(\cdot)$ is the variational estimator, which is only used for centralized training. We define the opposite of the lower bound as future predictor loss $\mathcal{L}_{FP}$. The derivation is defered to Appendix D.

In addition, we expect the personal message to help discard the irrelevant information while retain the future trajectory's information, which can endow the agent with a small state representation and accelerate training. We take advantage of the conditional entropy bottleneck (CEB) objective [5]. We maximize the mutual information between the personal message $\zeta_t^i$ and the future trajectory $\tau_t^i$, and minimize the mutual information between the current observation $o_t^i$ and the personal message $\zeta_t^i$ conditioned on the future trajectory $\tau_t^i$ simultaneously. The objective has the following lower bound:

$$I(\tau_t^i; \zeta_t^i) - I(o_t^i; \zeta_t^i | \tau_t^i) \geq \mathbb{E}_{o_t^i, \zeta_t^i, \tau_t^i} \left[ \log \frac{q_\phi(\zeta_t^i | o_t^i, \tau_t^i)}{\frac{1}{K} \sum_{j=1}^K q_\phi(\zeta_t^j | o_t^j, \tau_t^j)} - \log \frac{f_\psi(\zeta_t^i | o_t^i)}{q_\phi(\zeta_t^i | o_t^i, \tau_t^i)} \right], \tag{4}$$

where $f_\psi(\cdot)$ is the encoder for personal message, and $K$ is the size of the mini-batch sampled for training. We define the opposite of the lower bound as $\mathcal{L}_{CEB}$. See Appendix D for detailed derivation.

### 4.2.2 Conductor Election (CE)

In this section, we introduce three conductor election mechanisms: Random CE, Determinantal Point Process (DPP) [22] based CE, and Policy Gradient [37] based CE. Note that CE differs from the concept of leader election (LE) [39] in distributed system. We elect conductors to obtain better agent cooperation for maximizing the team reward, and we assume a perfect communication channel. LE is usually deployed in real multi-agent systems with the probability of agent disconnection, with the target to maintain the cognitive consistency among agents.

**Random CE.** Each agent is elected as a conductor with an independent and identical probability $p_l$, which is fully decentralized. The expectation of the number of groups can be controlled by $p_l$. Since all group members need to communicate with their neighbors only for message passing, SOG with random CE doesn't need centralized commander during execution. Therefore, it can be parallelized easily during execution.

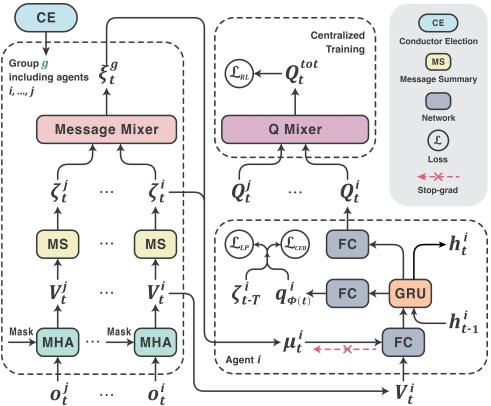

Figure 3: A diagram of the model structure. "MHA" means multi-head attention. We use this module to deal with the entity-wise input for dynamic team composition, detailed in Appendix A. $h_t^i$ is the hidden state for GRU cell.

**DPP-based CE.** In this part we expect to increase the generalization ability of SOG by maximizing the intra-group diversity. We achieve this by maximizing the diversity between conductors considering the conductor is the structural center of a group. Specifically, we formalize the conductor election as a determinantal point process $\mathbb{P}$.

**Definition 1.** *A determinantal point process (DPP) $\mathbb{P}$ is a probability distribution defined on the power set $2^{\mathcal{Y}}$ of a discrete finite basic point set $\mathcal{Y} = \{1, 2, ..., n\}$. $L \in \mathbb{R}^{n \times n}$ measures the similarity between any point pairs in $\mathcal{Y}$. Let $B$ be a random subset drawn by $\mathbb{P}$, then $\forall C \subseteq \mathcal{Y}$, we have*

$$\mathbb{P}(B = C) = \frac{\det(L_C)}{\det(L + \mathbf{I})}, \tag{5}$$

*where $L_C := [L_{i,j}]_{i,j \in C}$ is the submatrix of $L$, whose entries are indexed by the elements in $C$, and $\mathbf{I}$ is an $n \times n$ identity matrix. $\det(\cdot)$ means the determinant of a matrix.*

In practice, we use all agents to construct the point set $\mathcal{Y}$. Correlations are always non-positive in DPP [15]. The more similar two points are, the less possible they will appear in a subset sampled by DPP. Therefore, once we need to elect conductors, we just sample a subset $B$ by DPP and use all elements in $B$ as the conductors, which greatly reduces the probability to elect conductors with similar observations (e.g., two agents close to each other). For the similarity matrix $L$, we use the cosine similarity of agents' personal feature $V^i$:

$$L = [V^{1T}, V^{2T}, ..., V^{nT}]^T [V^1, V^2, ..., V^n] = [V^{iT}V^j]_{i,j \in n} \tag{6}$$

Another concern for DPP is the high computational cost for calculating the determinant of all submatrices. Instead of the traditional sampler by Schmidt orthogonalization [47], we find a method to calculate all determinants in parallel with a GPU, which greatly accelerates the computation. Details can be found in Appendix B.

**PG-based CE**. The conductor election might affect the group's accumulative rewards in the following time-steps. Therefore, we regard the conductor election task as a reinforcement learning problem. We use a policy $\boldsymbol{\pi}(\mathbf{a}|s) = \prod_{i=1}^{n} \pi^i(a|s)$ with the input of global state to decide the probability of each agent to become a conductor, and do policy gradient on it. The gradient $\nabla \mathcal{L}_{CERL}$ can be written as:

$$\mathbb{E}_{s_1, \mathbf{a}_1, ..., s_\infty, \mathbf{a}_\infty} \left[ \left( \sum_t r(s_t, \mathbf{a}_t) \right) \sum_{\hat{t}} \nabla \log \boldsymbol{\pi}(\mathbf{a}_{\hat{t}} | s_{\hat{t}}) \right], \tag{7}$$

where $\hat{t}$ is the time that CE happens, i.e., every $T$ time-steps. We omit the $\boldsymbol{\pi}$'s dependence on the parameter $\omega$ of the neural network for brevity. Electing conductors by PG takes the long-term benefits of the selection into consideration, and in experiments it shows great generalization ability.

### 4.3 Algorithm Outline

As shown in Fig. 3, our model is constructed using centralized training with decentralized execution paradigm. We utilize the entity-based input like in Iqbal et al. [12], so that the model can deal with the dynamic number of agents. The partial observability of local agents is assured by masking out the unseen entities. All agents share the same parameters. It's worth noting that the variational estimator $q_\phi$ is designed to include information of future $T$-steps. Since at time-step $t$ it takes the output of GRU cell $h_t^i$ as input, $q_\phi$ should be used to calculate the loss with message $\zeta$ before $T$ steps, i.e. $\zeta_{t-T}^i$. The local Q function uses the message $\xi^j$ sent by the conductor as part of input if it is in group $j$ when communication takes place, and uses personal message $\zeta^i$ in place at other time. Let $\boldsymbol{\mu}_t$ denote the set of the messages used for local Q prediction at time-step $t$. The mean square Bellman error objective for Q-learning is as follows:

$$\mathcal{L}_{RL}(\theta) = \mathbb{E}_{(\boldsymbol{\mu}_t, \boldsymbol{\mu}_{t+1}, \mathbf{a}_t, r_t, s_t, s_{t+1}) \sim \mathcal{D}} \left[ \left( r_t + \gamma \max_{\mathbf{a}'} Q_{\hat{\theta}}^{tot}(\boldsymbol{\tau}_{t+1}, \mathbf{a}' | \boldsymbol{\mu}_{t+1}) - Q_\theta^{tot}(\boldsymbol{\tau}_t, \mathbf{a}_t | \boldsymbol{\mu}_t) \right)^2 \right], \tag{8}$$

where $\hat{\theta}$ is the parameters of the target network, and $\mathcal{D}$ is the replay buffer. The overall loss can be written as:

$$\mathcal{L}_{all} = \mathcal{L}_{RL} + \lambda_1 \mathcal{L}_{FP} + \lambda_2 \mathcal{L}_{CEB}, \tag{9}$$

where $\lambda_1$ and $\lambda_2$ are hyper-parameters. We summarize the training procedure in Algorithm 1.

## 5 Experiments

In this section we design experiments to answer the following questions: (1) Whether the Self-Organized Group helps the agents coordinate better than fully-connected communication methods?

---

**Algorithm 1** Self-Organized Group

---

Initialize $\theta, \phi, \psi$. Set learning rate $\leftarrow \eta$, communication interval $\leftarrow T$, $\mathcal{D} \leftarrow \{\}, \lambda_1, \lambda_2, t_{max}$
**for** each episode iteration **do**
    **for** $t = 1, 2, \ldots, t_{max}$ **do**
        **if** $t \mod T \neq 0$ **then**
            For each agent $i$, $\mu_t^i \leftarrow Sample(f_\psi(o_t^i))$
        **else**
            Elect conductors and form groups $\{G_j\}_{j=1}^K$
            For each agent $i$, $\zeta_t^i \leftarrow Sample(f_\psi(o_t^i))$
            For each conductor $j$, $\xi_t^j \leftarrow Mix(\zeta_t^i), i \in G(j)$
            For each agent $i$, $\mu_t^i \leftarrow \xi_t^j, i \in G(j)$
        **end if**
        Generate tuple$\{\mathbf{s}_t, \mathbf{a_t}, r_t, \mathbf{s}_{t+1}, \boldsymbol{\zeta}_t, \boldsymbol{\zeta}_{t+1}\}$ by executing $f_\theta(\boldsymbol{\tau}_t, \boldsymbol{\mu}_t)$
    **end for**
    $\mathcal{D} \leftarrow \mathcal{D} \cup \{\mathbf{s}_t, \mathbf{a_t}, r_t, \mathbf{s}_{t+1}, \boldsymbol{\zeta}_t, \boldsymbol{\zeta}_{t+1}\}_{t=1}^{t_{max}}$ , $\theta \leftarrow \theta + \eta \hat{\nabla}_\theta \mathcal{L}_{RL}(\mathcal{D})$
    $\psi \leftarrow \psi + \eta(\hat{\nabla}_\psi \mathcal{L}_{FP}(\mathcal{D}) + \hat{\nabla}_\psi \mathcal{L}_{CEB}(\mathcal{D}))$ , $\phi \leftarrow \phi + \eta(\hat{\nabla}_\phi \mathcal{L}_{FP}(\mathcal{D}) + \hat{\nabla}_\phi \mathcal{L}_{CEB}(\mathcal{D}))$
**end for**

---

(2) Can Self-Organized Group promote zero-shot generalization to the dynamic team composition and the varying partial observability, even on complicated tasks? (3) What's the effect of the conductor election way? (4) What factor contributes to Self-Organized Group most? We test our idea on three commonly used multi-agent benchmarks: Resource Collection, Predator-Prey, and StarCraft II micromanagement tasks. The previous two are built on the multi-agent particle environment [21], and the last is modified from Iqbal et al. [12]. The number of the agents is not fixed, sampling from a pre-defined set at the beginning of each episode in all scenarios. For all the three tasks the testing environments are more complicated than the training ones. Agents need to handle the situations that is not encountered during training. Each experiment is repeated 3 or 5 times with different seeds.

In experiments, the name "SOG" means our method with random CE, whose execution is decentralized. "SOG_dpp" and "SOG_rl" means the SOG with DPP-based CE and PG-based CE, respectively. Except for our method, we evaluate four state-of-the-art CTDE methods that are suitable for dynamic team composition: A-QMIX [46], REFIL [12], EMP [1] and COPA [18]. For a fair comparison, we extend the original EMP's decentralized-training structure to CTDE. For A-QMIX and REFIL, agents have no communication during execution, while for EMP and COPA, agents communicate with their neighbors or a global coach. Since many MARL methods with communication are not suitable for dynamic team composition (e.g., CommsNet [34] and G2ANet [19]), we extract one scenario of Resource Collection and show results in Appendix G.1. We also compare SOG with graph-based methods including NCC [23], MAGIC [26] and Gated-ACML [24] in Appendix I.

## 5.1 Resource Collection

This scenario is modified from the environment described in Liu et al. [18]. Agents need to collect resources from 6 resource points and transport the goods home. The number of the agents for training is uniformly sampled from $\{2,3,4,5\}$, while for testing it is sampled from $\{6,7,8\}$. Each agent has a sight range $SR$. Entities including other agents and resource points that exceed agent $i$'s $SR$ are invisible to agent $i$. For more details of the environment, please refer to Appendix F.1.

We train on two kinds of $SR$: 0.5 and 1.0. We show in Fig. 4(a-b) the results of all methods. SOG significantly outperforms the other methods when $SR = 0.5$ and $SR = 1.0$. Given that the testing scenario includes 6/7/8 agents while the training scenario only has 2/3/4/5, the communication mechanism of SOG greatly strengthens SOG's zero-shot generalization ability for dynamic team composition. The CE methods have no obvious difference when the team size varies.

Then we test the zero-shot generalization ability in different environmental conditions. We save the previously trained models with a fixed sight range 0.5 or 1.0 and evaluate them on different agent sight range settings. Each setting is repeated for 160 testing episodes and the results are averaged over 3 models with different seeds. We show the performance of each method in Fig. 4(c-d). Except for the difficult setting $SR = 0.2$ that all methods have poor performance, SOG and SOG_rl can achieve

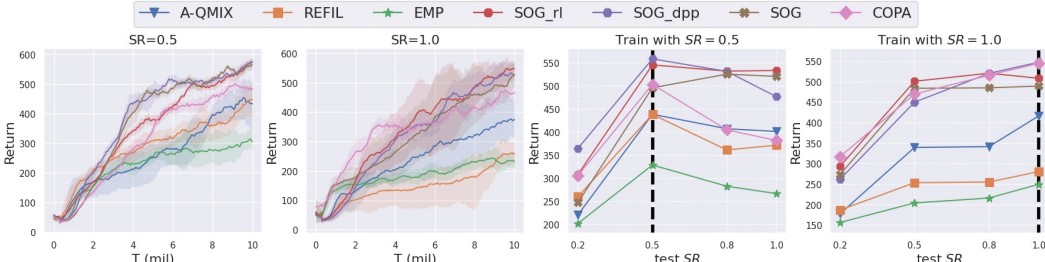

Figure 4: (a-b) The average episode returns in the scenario Resource Collection with 2 kinds of sight ranges (SR). (c-d) Their performance on different test sight ranges from $SR = 0.2$ to $SR = 1.0$.

similar or even better performance in different scenarios ranging from $SR = 0.5$ to $SR = 1.0$ than in training scenarios. SOG_rl even performs better transferring from $SR = 1.0$ to $SR = 0.8$. In the meantime, other methods show notable performance drop when transferred from training settings to others. The results show SOG's strong adaptation to unseen scenarios.

## 5.2 StarCraft Micromanagement Tasks

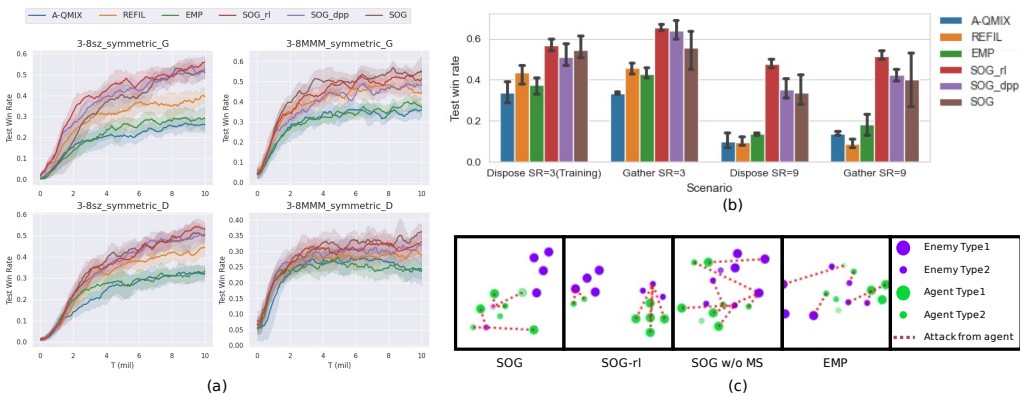

Figure 5: (a) Results of the test winning rates on StarCraft2. (b) Models' performance when transferred to unseen scenarios. (c) Visualization of agents' behaviour at time step 20.

We apply our method to the StarCraft multi-agent challenge (SMAC) [30]. We use the map designed by Iqbal et al. [12] and Liu et al. [18]. The maps randomly initialize 3-8 agents with enemies of the same number at the start of each episode. Agents are initialized together with the sight range 9 [12] or divided into 2-4 groups with the sight range 3 [18]. We call the previous one "G(ather)" and the latter "D(isperse)". In the scenario "Disperse", the enemies are divided into 1-2 groups, which means each group of enemies is stronger than that of the agents. We limit the number of agents to 3-5 in training procedure and test the model with agent number 6-8. Similar to Iqbal et al. [12], Liu et al. [18], we also incorporate the imaginary objective.

We first explore the adaptation to the dynamic team composition. We show in Fig. 5(a) the results on 4 maps. Our method has the highest test winning rate against the preset AI when testing the model with a larger size of agents and enemies than training. And the PG-based CE improves the performance on 3-8sz. Combining the results in Resource Collection, we conclude that PG-based CE can improve agents' zero-shot generalization ability on dynamic team composition in many cases, due to the more reasonable conductor election mechanism than random CE.

Then, we study the adaptation to the varying environment conditions of our method. We save the models trained on map 3-8sz_D with the sight range 3, which is a hard case that agents are split and have restricted horizons. Then we evaluate the models on other initial conditions and sight ranges. The results are shown on Fig. 5(b). The initial conditions (Gather or Disperse) have no obvious effect on the performance for all four methods. However, when agents can see more entities than training,

other three methods all show large performance drop while our method has comparable performance to that in training. This is probably due to the unified messaging passing mechanism within the group. An interesting phenomenon is that the random CE has comparable or better performance to rl-based CE on 2 scenarios. We speculate this is due to the similar pattern of random CE in different environment settings, and analyse this result in detail in Appendix G.2. The results show that our method can better adapt to the varying partial observability.

To further validate the effect of our method, we visualize the situation for four methods (SOG, SOG-rl, SOG without MS, and EMP3) on the same SMAC map in Fig. 5(c). We can see that agents trained by SOG learn to gather and concentrate fire on certain enemies surrounded by agents. SOG-rl sacrifices a group of 2 agents to exchange the the elimination of 3 enemies. SOG without message summary learns to gather, but the target of each agent's attack is not focused. EMP3 does not even learn to attack simultaneously. The results show that CE and MS both promote better agent cooperation.

### 5.3 Predator Prey

In this scenario, agents play the role of predators, aiming to catch some preys with random walk. The training number of predators is sampled from {3,4}, along with only 1 prey. During execution we initialize 5 or 6 predators, as well as 1 or 2 preys. For environment details, please refer to Appendix F.2. It is a relatively easy-to-learn scenario, and we make ablations on it.

First, we explore the effect of the message summarizer. Except for the $\mathcal{L}_{FP}$ and $\mathcal{L}_{CEB}$, we also try an entropy regularizer similar to the implementation of Wang et al. [42]. It aims to maximize the entropy of the message to encourage exploration. As shown in Table 1, the combination of $\mathcal{L}_{FP}$ and $\mathcal{L}_{CEB}$ obtains the best average testing return, exceeding the performance of CTDE method A-QMIX, and communication method EMP. Although the entropy regularizer helps stabilize the training (the variance is reduced when combined with the regularizer), it makes no contribution to the average performance. Therefore we use the $\mathcal{L}_{FP} + \mathcal{L}_{CEB}$ as our default loss for message summarizer. Notice that EMP has similar performance to SOG on training scenarios. Nevertheless, SOG performs better when transferred to complicated evaluation scenarios.

| Method | $\mathcal{L}_{FP}$ | $\mathcal{L}_{CEB}$ | Reg | SR | CR |
|---|---|---|---|---|---|
| EMP | | | | 9±0.62 | 9.37±2.32 |
| A-QMIX | | | | 8.03±1.4 | 7.57±2.44 |
| | ✓ | | | 7.95±0.61 | 7.2±1.15 |
| | | ✓ | | 8.21±0.7 | 7.53±1.72 |
| SOG | ✓ | ✓ | | 9.10±0.56 | 9.98±1.15 |
| | ✓ | ✓ | ✓ | 7.79±0.09 | 7.31±0.08 |

Table 1: The results of different losses on Predator-Prey. "Reg" means the entropy regularizer. "SR" is the average return of 160 testing episodes on simple training scenarios, while "CR" is the counterpart on complicated evaluation scenarios.

Then we analyze the choice related to communication in Table 2, including the number of groups in expectation, the message dimension, and the communication interval. The number of group 2 has better performance than number 1 or 4. Since the rule that a prey needs to be caught by 3 predators simultaneously, it is suitable to divide 5 or 6 agents into 2 groups. When the message dimension is reduced from 10 to 3, the results have no obvious drop. However, it decrease greatly when the message dimension is set to 1. The results show that a 3-dimensional message is sufficient for communication for Predator-Prey task. As for communication interval, $T = 2$ is a little better than $T = 4$, and $T = 10$ is not enough for sufficient communication.

| Group Num | 1 | 2(D) | 4 |
|---|---|---|---|
| Test Return | 7.43±1.88 | 9.90±3.20 | 7.37±3.79 |

| Msg dim | 1 | 3(D) | 10 |
|---|---|---|---|
| Test Return | 6.30±2.0 | 9.90±3.20 | 10.77±0.91 |

| T | 2 | 4(D) | 10 |
|---|---|---|---|
| Test Return | 10±0.79 | 9.90±3.20 | 7.5±1.05 |

Table 2: Ablation studies. "(D)" means the default setting.

## 6 Discussion on Limitations

As stated in the section 4.2.2, we only test SOG on the perfect communication channel. Its performance may fall down when faced with broken communication channel. Another limitation for random CE is that the expectation of the size of each group is decided by a pre-defined hyperparameter, i.e., the probability of agent elected as a conductor. When transferred to unseen scenarios, if the size of each group differs a lot from the training condition, it may cause the performance drop. We try to git rid of the hyperparameter by introducing DPP-based CE and RL-based CE, but they both require a centralized conductor elector, which is not full CTDE.

# 7 Negative Societal Impact

There is a possibility that our method could be employed in real-world multi-agent systems such as UAV formation or intelligent warehouse management. Utilizing the policy our method derives directly is risky, since there is a domain gap between the training virtual environment and real-world scenarios. The practitioners are supposed to restrict their policy under human supervision to avoid harmful options.

# 8 Conclusion

In this paper we propose Self-Organized Group (SOG) for cooperative multi-agent reinforcement learning. In SOG, a certain number of agents are randomly elected to be conductors and the corresponding groups are constructed with conductor-follower consensus, allowing the groups to be re-organized every $T$ time-steps. We find the organized group under the unified command of a conductor embeds the multi-agent system with stronger zero-shot generalization ability compared to the traditional CTDE methods with fully-connected communication mechanism. Furthermore, we derive a variational message summarizer for efficient and economical message passing, and we propose DPP-based and PG-based conductor election strategies for better group organization. We take experiments on three commonly used multi-agent benchmarks. SOG shows better zero-shot generalization ability not only for the dynamic team composition, but also for the varying partial observability on all three benchmarks.

# 9 Acknowledgement

This work was supported by the National Key R&D Program of China under Grant 2018AAA0102801, National Natural Science Foundation of China under Grant 61620106005.

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
