# A    Entity-wise Input

It's critical to deal with the dynamic team composition in real-world multi-agent scenarios. However, traditional methods usually fix the team size when the network is built. A common solution is to introduce a multi-head attention ($MHA$). To feed MHA, the raw state $s$ of the environment is represented as a series of entities $e^i$, i.e., $s^e := \{e^i\}, i \in [1, n_e]$ with the same vector length, where $n_e$ is the maximum number of entities. The entities include agents we can or can not control, and other substrates in the multi-agent scenario (e.g. obstacles). After some linear layers, $s^e$ is updated as a matrix $\mathcal{X} \in \mathbb{R}^{n_e \times d}$ where $d$ is the dimensionality of the entity representation. Each agent's partial observability $o^i$ can be obtained by a mask $m$ with size $n_e$ multiplied on $s^e$. Elements in $m$ equal one if the corresponding entity is in the agent's sight range else 0. Combining all agents' masks together, the matrix $\mathcal{M} \in \mathbb{R}^{|I| \times n_e}$ indicating all agents' observability, where $I$ is the agent set. An MHA module, denoted as $MHA(I, \mathcal{X}, \mathcal{M})$, can integrate information not blocked by masks across all entities. Taking only global entities as inputs during the decentralized execution, CTDE with MHA can easily achieve agents' partial observability with entity-wise expression. Compared to the traditional agent-wise customized observation which may bring information redundancy, entity-wise input is concise and efficient. For detailed structure of the $MHA$ module, please refer to Iqbal et al. [2].

# B    Implementation of the Fast DPP Sampler

If we want to sample from a DPP, we need to calculate the determinant of all submatrices of the similarity matrix $L \in \mathbb{R}^{n \times n}$. We show in Alg. 1 the procedure.

---

**Algorithm 1** Pseudo code of Fast DPP Sampler in a PyTorch-like style.

```
# n: number of agents.
# cos_matrix: the similarity matrix with shape [batch_size, n, n]

# Generate all submatrices with shape n×n. Fill the not-in-submatrix place with 0.
mask = 2 ** torch.arange(n - 1, -1, -1)
x = torch.arange(2 ** n)
y = x.unsqueeze(-1).bitwise_and(mask).ne(0)
sub_matrix_mask = torch.logical_and(y.unsqueeze(1), y.unsqueeze(-1))
cofactor_matrix = cos_matrix.unsqueeze(1) * sub_matrix_mask.unsqueeze(0) # shape [batch_size,
2^n, n, n]

#  Fill the positions in the diagonal that are not in the submatrix with 1.
missed_ones = torch.diag(torch.ones(n)).unsqueeze(0)*torch.logical_not(sub_matrix_mask)
cofactor_matrix += missed_ones.unsqueeze(0)

# Calculate the determinants in parallel.
p = torch.linalg.det(cofactor_matrix)

# Sample the conductors.
cat = Categorical(probs=p)
conductors = cat.sample()

# Get a mask of shape [batch_size, n]. 1 means a conductor, and 0 means a follower.
conductor_mask = conductors.unsqueeze(-1).bitwise_and(mask).ne(0)
```

---

# C    Code and Reproducibility

We commit our code in `https://github.com/thu-rllab/SOG`.

## D  Derivation of Message Summarizer

First we give the derivation of $\mathcal{L}_{FP}$. To simplify the notation, we omit the superscript $i$ and the subscript $t$, since the message is summarized by any agent $i$ at any time-step $t$. We fully derive the Equation. (3) as follows:

$$
\begin{aligned}
I(\zeta;\tau|o) &= \mathbb{E}_{o,\zeta,\tau}\left[\log\frac{p(\zeta|\tau,o)}{p(\zeta|o)}\right]\\
&= \mathbb{E}_{o,\zeta,\tau}\left[\log\frac{q_\phi(\zeta|\tau,o)}{p(\zeta|o)}\right] + KL\left(p(\zeta|\tau,o), q_\phi(\zeta|\tau,o)\right)\\
&\geq \mathbb{E}_{o,\zeta,\tau}\left[\log\frac{q_\phi(\zeta|\tau,o)}{p(\zeta|o)}\right]\\
&= \mathbb{E}_{o,\zeta,\tau}\left[\log q_\phi(\zeta|o,\tau)\right] + H(\zeta|o),
\end{aligned}
\tag{1}
$$

The inequality holds because any $KL(\cdot,\cdot) \geq 0$.

Then we derive $\mathcal{L}_{CEB}$. For simplicity we omit the subscript $t$. Recall that we want to maxamize $-I(o_t^i;\zeta^i|\tau^i) + I(\tau^i;\zeta^i)$. The first term can be written as:

$$
\begin{aligned}
-I(o_t^i;\zeta^i|\tau^i) &= \mathbb{E}_{o^i,\zeta^i,\tau^i}\left[-\log\frac{p(\zeta^i|o^i,\tau^i)}{p(\zeta^i|\tau^i)}\right]\\
&= \mathbb{E}_{o^i,\zeta^i,\tau^i}\left[\log\frac{p(\zeta^i|\tau^i)}{p(\zeta^i|o^i,\tau^i)}\right]\\
&= \mathbb{E}_{o^i,\zeta^i,\tau^i}\left[\log\frac{q_\phi(\zeta^i|\tau^i)}{p(\zeta^i|o^i,\tau^i)}\right] + KL(p(\zeta^i|\tau^i), q_\phi(\zeta^i|\tau^i))\\
&\geq \mathbb{E}_{o^i,\zeta^i,\tau^i}\left[\log\frac{q_\phi(\zeta^i|\tau^i)}{p(\zeta^i|o^i,\tau^i)}\right]
\end{aligned}
\tag{2}
$$

Since the message is summarized before the future trajectories are generated, the message $\zeta^i$ is independent of the future trajectory $\tau^i$. Therefore we can use a neural network parameterized by $\psi$, which only takes $o^i$ as the input, to represent the message summarizer, i.e., we can use $f_\psi(\zeta^i|o^i)$ to represent $p(\zeta^i|o^i,\tau^i)$. To get a variational lower bound of the second term $I(\tau^i;\zeta^i)$, we use the CatGen formulation, i.e., Equation.(5) in Lee et al. [3]:

$$
I(\tau^i;\zeta^i) \geq \mathbb{E}_{\zeta^i,\tau^i}\left[\log\frac{p(\zeta^i|\tau^i)}{\frac{1}{K}\sum_{j=1}^K p(\zeta^j|\tau^j)}\right],
\tag{3}
$$

where we estimate the mutual information $I(\tau^i;\zeta^i)$ for a single example from a minibatch with size $K$, and the $K$ messages are sampled independently. The future trajectories $\tau^i$ depends on the current observation $o^i$, so we can reuse the variational encoder $q_\phi(\zeta^i|o^i,\tau^i)$ to represent $p(\zeta^i|\tau^i)$. Therefore, we can write the whole CEB objective as follows:

$$
I(\tau^i;\zeta^i) - I(o_t^i;\zeta^i|\tau^i) \geq \mathbb{E}_{o^i,\zeta^i,\tau^i}\left[\log\frac{q_\phi(\zeta^i|o^i,\tau^i)}{\frac{1}{K}\sum_{j=1}^K q_\phi(\zeta^j|o^j,\tau^j)} - \log\frac{f_\psi(\zeta^i|o^i)}{q_\phi(\zeta^i|o^i,\tau^i)}\right].
\tag{4}
$$

## E  Hardware and Hyperparameters

We use an AMD Ryzen 3975WX CPU with 32-Cores and three RTX-3090-11G GPUs to run all the experiments with three seeds. One experiment of Resource collection and Predator-prey takes around 10 hours. For StarCraft Micromanagement Tasks, it takes about 20 hours to run one experiment.

We summarize some hyper-parameters in Table. 1:

## F  Detailed Environment Description

### F.1  Resource Collection

In Resource Collection, agents need to collect resources from 6 resource points and transport the goods home in a map of boundary [-1,1]. The location of the resource point and the home are random

| Name | Description | Value |
|------|-------------|-------|
| $\gamma$ | Discounted factor | 0.99 |
| $\varepsilon$ anneal time | Time-steps for $\varepsilon$ to anneal from $\varepsilon_s$ to $\varepsilon_f$. $\varepsilon$ is the probability for agents choosing random actions. | 500000 |
| $\varepsilon_s$ | Start $\varepsilon$ | 1 |
| $\varepsilon_f$ | Final $\varepsilon$ | 0.05 |
| $n_{env}$ | The number of parallel environments | 8 |
| $|\mathcal{D}|$ | Replay buffer size | 5000 |
| $n_{head}$ | Number of heads in multi-head attention | 4 |
| $n_{attn}$ | Dimension of Attention embeding | 128 |
| $n_{rnn}$ | Dimension of RNN cells | 64 |
| $lr$ | Learning rate | 0.0005 |
| $\alpha$ | $\alpha$ value in RMSprop | 0.99 |
| $\epsilon$ | $\epsilon$ value in RMSprop | 0.00001 |
| $n_{batch}$ | Batch size | 32 |
| $t_{target}$ | Time interval for updating the target network | 200 |
| $\lambda_1$ | Weight of $\mathcal{L}_{FP}$ | 0.01 |
| $\lambda_2$ | Weight of $\mathcal{L}_{CEB}$ | 0.01 |
| $G_{max}$ | Clipping value for all gradients | 10 |

Table 1: Hyper-parameters.

places uniformly sampled from the whole map. The radius of the home and the resource location & agent is 0.1 and 0.05, respectively. There are 3 kinds of resources, and each agent $i$ has its own ability $b_i^e$ uniformly sampled from $\{0.1, 0.5, 0.9\}$ to collect each kind of resource $e$. Each agent can accelerate towards 4 directions or apply no forces at each time-step. Each agent has its maximal speed uniformly sampled from $\{0.3, 0.5, 0.7\}$ and the acceleration is fixed to 3.0. Every time the agent $i$ collects resource $e$, the team will get a reward $10 * b_i^e$. When an agent brings the resource home, the team will get a reward 1. An agent can only carry one resource at a time, which means an agent needs to bring the collected resource home before it starts to collect the next one. The episode limit is 145. The number of the agents for training is uniformly sampled from $\{2,3,4,5\}$, while for testing it is sampled from $\{6,7,8\}$. Each agent has a sight range $SR$. Entities including other agents and resource points that exceed agent $i$'s $SR$ are invisible to agent $i$.

## F.2  Predator Prey

In this scenario, agents play the role of predators, aiming to catch some preys with random walk in a map of boundary [-1,1]. The acceleration for predator/prey is 3.0/4.5. The maximal speed for predator is uniformly sampled from $\{0.2, 0.3, 0.5\}$, while it is $\{1.0, 1.2\}$ for prey. At each episode there are 3 obstacles with radius 0.05 generated, together with 2 tunnels. A prey stepping into one tunnel will be immediately transported to the other. Agents need to learn to cooperate and guard at the exit of the two tunnels to catch the preys. Only when at least 3 predators touch the prey simultaneously, the prey is caught. The team will get the reward 25 for catching one prey, and is penalized for $-0.1 * \sum_j \min_i D_{ij}$ at each time step. $D_{ij}$ is the distance of agent $i$ and prey $j$. The episode ends if all preys are caught, or it reaches the episode time limit. The training number of predators is sampled from $\{3,4\}$, along with only 1 prey. During execution we initialize 5 or 6 predators, as well as 1 or 2 preys.

# G  More Results on Methods with Fixed Agent Input and Heuristic Conductor Election.

## G.1  Fixed Agent Input

Many communication baselines can only handle a fixed number of agents. Therefore, it is difficult to apply them to our environments with dynamic team composition. To make a fair comparison with them, we fix the agent number in Resource Collection to 6 and run 3 seeds for CommsNet [9], G2ANet[5] and COPA [4] separately. For COPA we also run it on the dynamic team composition in

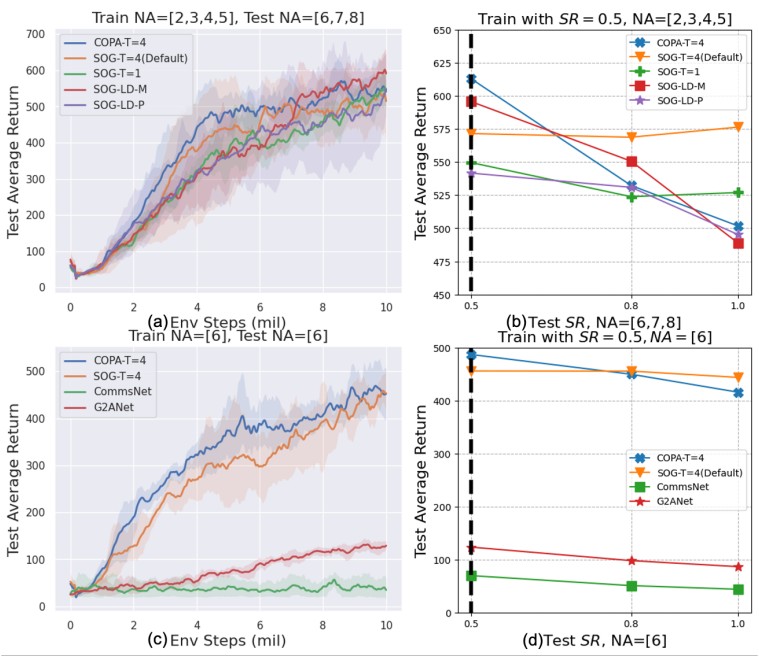

Figure 1: Additional results on fixed agent number and heuristic conductor election.

Fig. 1(a). We show in Fig. 1(c) the results. On the fixed environment, COPA performs a little better than SOG, while CommsNet and G2ANet almost fail to learn effective policy. We also test the trained model on different sight ranges in Fig. 1(d). SOG shows better adaptation to unseen sight ranges than COPA. We emphasize that SOG is easier to distribute since it enjoys decentralized execution while COPA, CommsNet and G2ANet are not fully decentralized.

## G.2 Heuristic Conductor Election

The main body shows that random conductor election has a little better zero-shot generalization ability than DPP and policy gradient in some scenarios. In Fig. 1(a,b), we also add two kinds of heuristic conductor election mechanisms in comparison. The first (SOG-LD-M) elects conductors with the maximal neighbor numbers, while in the second manner (SOG-LD-P), the probabilities of agents elected as conductors are proportional to their neighbor numbers. We can see that SOG-LD-M performs a little better for the unseen agent numbers, while both heuristic conductors (and COPA, which has a central conductor) show obvious performance drop in unseen sight ranges. Possible reasons are as follow: Although in fixed environments, a delicate conductor election can obviously improve the agents' performance, we propose that random conductor election can benefit the agents' zero-shot generalization ability. Agents may face more combinations of different conductors and team members with random conductor election, which can improve their ability and robustness to deal with unseen scenarios. It can be regarded as a structural exploration on agents' relationship, which may find better coordination style than heuristic ways. Similar ideas are proposed in REFIL [2]. They randomly masked out some teammates to deal with dynamic team composition.

## H Quantitative Analysis on the Communication Cost

Assume the number of agents is $N$, and the average number of each agent's neighbors is $M$. The number of messages delivered at each time-step for fully-connected communication like Agarwal et al. [1] is $O(MN)$, since each agent is required to send messages to all its neighbors. By contrast, in our proposed method, the overall cost is just $O(N)$. The proof is as follows: We divide the messages into two types: sent by conductors and sent by followers. For followers, each agent only needs to send one message to its conductor, therefore the communication cost is $O(N)$. For conductors, each follower is only subject to one conductor, so each agent can only receive at most one message sent by

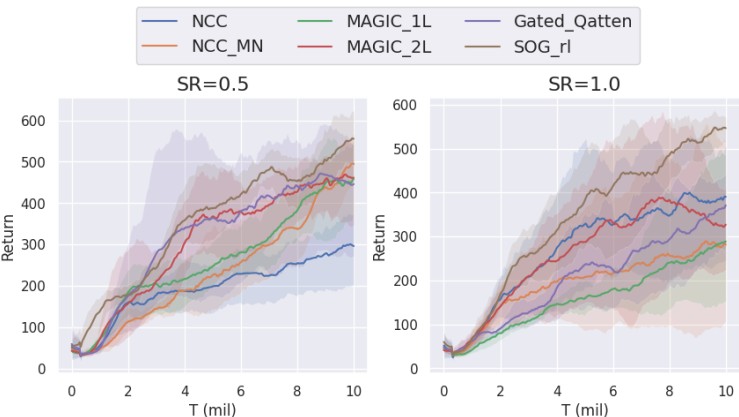

Figure 2: Comparison to NCC [6], Gated-Qatten [7] and MAGIC [8]. "NCC_MN" means the NCC with multi-neighborhood implementation and "NCC" means the single neighborhood version. "MAGIC_1L" means the 1-layer processor & scheduler version and "MAGIC_2L" is the 2-layer implementation. We find that the 2-layer MAGIC is better than the 1-layer one. The multi-neighborhood version of NCC is suitable for short sight range while the single-neighborhood is good for large SR. Gated_Qatten has similar performance to the 2-layer MAGIC. And SOG-rl shows better zero-shot generalization ability.

conductors, costing $O(N)$. Therefore the overall messages delivered at one time-step is $O(N)$ in our proposed method.

# I    Comparison to Attention-based Methods

Figure 2 shows a comparison of SOG-rl, NCC [6], Gated_Qatten [7] and MAGIC [8]. All methods are re-implemented to adapt to our environment with dynamic team composition. By keeping the core part unchanged, we apply all these methods to the value-based framework to make a fair comparison.

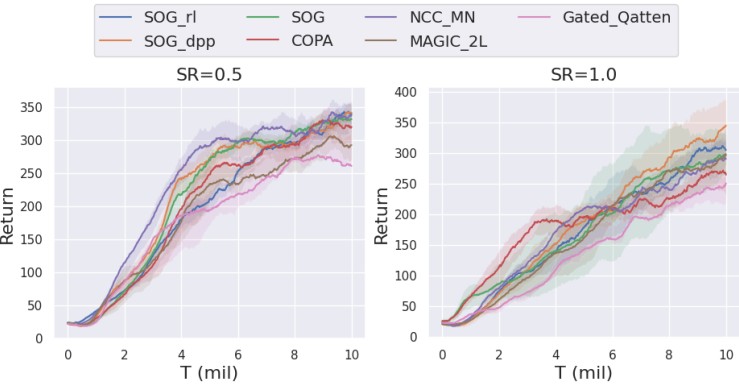

Figure 3: Training curves on Resource Collection of our SOG, COPA [4], NCC [6], Gated-Qatten [7] and MAGIC [8].

# J    Learning Curves for Training Performance

To test whether SOG's advantage is already visible during training or specific to its generalisation ability, we plot the training return curves in Resource Collection (agent number sampled in [2,3,4,5]) in Fig 3. To make the curve more clear, we do not show algorithms with relatively low returns, including QMIX_atten, EMP and REFIL. The results show that SOG has no obvious superior

performance than NCC or MAGIC in training, which indicates that the advantage of SOG is specific to its generalisation ability. Next we give a detailed discussion about the source of generalization ability.

The generalization ability mainly comes from the time-varying group communication mechanism. Many existing proximity-based methods regard the communication between agents as a graph. When transferred to unseen scenarios, the graph's degree and the number of edges may change a lot, i.e., the communication pattern changes a lot, thus causing the performance drop. In contrast, through our SOG mechanism, the agent can maintain a similar communication pattern to the training condition. For example, when training a model with 2-agent team to fulfill a task, it may perform worse in a 4-agent scenario, since the agent is only trained to cooperate with another one agent, and it may be confused by the message sent by the other 2 agents. However, our proposed SOG mechanism has a high probability to divide the 4-agent into 2 groups, and then preventing the message delivery between the two groups. By doing so, in unseen scenarios, the agent may find the communication pattern is similar to that in training, and perform a similar 2-agent coordination pattern. Therefore, an organized group under the unified command of a conductor can better adapt to an unseen scenario than individuals.