# OpenReview forum: "Self-Organized Group for Cooperative Multi-agent Reinforcement Learning"
_NeurIPS.cc/2022/Conference — NeurIPS 2022 Accept_

### Official Review · Reviewer_646o · 2022-07-06

**Rating:** 5
**Confidence:** 4
**Soundness:** 2 fair
**Presentation:** 3 good
**Contribution:** 2 fair

**Summary:**

This paper proposes a Self-Organized Group mechanism for Cooperative MARL, which is claimed to have zero-shot generalization ability with dynamic team composition and varying partial observability. The mechanism first selects a condutor for each group; then the condutor summarizes the messages (by conditional entropy bottleneck objective) from all agents within this group, and sends the summarized message to all agents; finally, the agents generate actions based on the observation and the summarized message. For the condutor selection, the authors proposed Random, DPP-based and Policy Gradient-based methods. The authors conducted experiments on three environments to show the effectiveness of the proposed methods.

**Questions:**

* What is the relation between the proposed Self-Organized Group mechanism (i.e., both Conductor Election and Message Summary) and the so claimed zero-shot generalization ability?
* Could the authors give some results by comparing with [1, 5] or other more strong baselines? Because [1] has a message prunning mechanism, which is like the T-timestep communication in the current paper. Similarly, the group-based multi-agent Message Summary is similar to the Neighborhood Cognition Consistent proposed in [5], and [5] can also be applied to dynamic team composition.

**Limitations:**

Yes

**Strengths And Weaknesses:**

Strengths
* Clear paper writting. In general, this paper is easy to read and follow.
* Interesting methods for Conductor Election.





Weaknesses
* The methods and the claims are not related. I do not see the direct relation between the proposed Self-Organized Group mechanism (i.e., both Conductor Election and Message Summary) and the so claimed zero-shot generalization ability with dynamic team composition and varying partial observability. In my opinion,  the zero-shot generalization ability is mostly because of the mean message mixer (i.e., Eq 1) and the MHA-based model structure (i.e., Fig 3).
* Missing a lot of related works. For example, the Conductor is similar to the centralizer/coordinator used in many MARL communication methods [1,2]. The entropy bottleneck based Message Summary is also related with entropy bottleneck based communication message generation [3, 4]. The group-based multi-agent Message Summary is also similar to the Neighborhood Cognition Consistent proposed in [5], where summarized message is equal to the consistent cognition. But the authors missed these works.
* Due to the above reason, I found the baselines used in the experiments are not strong. For example, the authors mentioned "Since many MARL methods with communication are not suitable for dynamic team composition", but as far as I know, [1, 4, 5] and many other MARL methods can be applied for dynamic team composition.


[1] Learning agent communication under limited bandwidth by message pruning. AAAI 2020.
[2] Learning multi-agent communication with double attentional deep reinforcement learning. AAMAS 2020.
[3] Deep multi agent reinforcement learning for autonomous driving. 2020.
[4] Multi-agent Communication with Graph Information Bottleneck under Limited Bandwidth. 2021.
[5] Neighborhood cognition consistent multi-agent reinforcement learning. 2020.

---

> ### Author Response · Authors · 2022-08-02
> **Response to Reviewer 646o**
>
> Thank you for your constructive feedback.
> ### Q1: Relation between SOG and zero-shot generalization ability.
> The generalization ability mainly comes from the time-varying group communication mechanism. Many existing proximity-based methods regard the communication between agents as a graph. When transferred to unseen scenarios, the graph's degree and the number of edges may change a lot, i.e., the communication pattern changes a lot, thus causing the performance drop. In contrast, through our SOG mechanism, the agent can maintain a similar communication pattern to the training condition. For example, when training a model with 2-agent team to fulfill a task, it may perform worse in a 4-agent scenario, since the agent is only trained to cooperate with another one agent, and it may be confused by the message sent by the other 2 agents. However, our proposed SOG mechanism has a high probability to divide the 4-agent into 2 groups, and then preventing the message delivery between the two groups. By doing so, in unseen scenarios, the agent may find the communication pattern is similar to that in training, and perform a similar 2-agent coordination pattern. Therefore, an organized group under the unified command of a conductor can better adapt to an unseen scenario than individuals.
>
> ### Q2: Missing related works.
> Since the related works [1,5] are either actor-critic or using fixed agent number, we can not directly apply these methods to the environment we used. Therefore, we have implemented the above two methods under our value-based framework with entity-wise input. ~~Due to our limited computing resources, we first show the results of [5] in Appendix J and will attach the results of [1] as soon as possible.~~ 8.8 update, we show the results of [1]\(Gated_Qatten\) and [5]\(NCC and NCC_MN\) in Appendix J. The multi-neighborhood version of NCC is suitable for short sight range while the single-neighborhood is good for large SR. Gated_Qatten shows relatively better performance than the 2 versions of NCC, and SOG-rl performs better on both. We will also give more detailed related works about [2,3,4] in the final version.
>
> ### Q3: The difference between our Conductor and previous works.
> In many previous works including [1,2], the conductor is usually designed as a module in end-to-end neural networks, rather than the agent itself. In contrast, we elect conductor from homogenous agents and wish to introduce heterogeneity to the agents. The advantage of our conductor is that it brings no extra computational cost when building a heterogeneous communication mechanism.
>
> ### Q4: The difference between our Message Summarizer and previous works.
> The message generation in [3,4] is designed for reducing the entropy, so that to communicate under the low-bandwidth limitation. However, the target of our Message Summarizer is discarding the unrelated information for predicting the future state. Therefore, our optimization objective is different from that in [3,4]. Since our testing environment assumes no bandwidth limitation, the results in Table 1 shows the effect of Message Summarizer on improving performance and zero-shot generalization ability.
>
> ### Q5: The difference between SOG and Neighborhood Cognition Consistent.
> Neighborhood Cognition Consistent[5] can be regarded as a soft constraint on the cognition of nearby agents, while SOG's communication mechanism is a hard constraint in the group. Another difference is that SOG has no constraint between groups, which we think is useful for zero-shot generalization ability. As we stated in Q1, NCC regards the constraint between agents as a graph, and the graph may change a lot when transferred to unseen scenarios, causing the performance drop. In contrast, SOG cuts off the connection between groups, and agents may find more familiar coordination pattern.

---

> > ### Comment · Reviewer_646o · 2022-08-09
> > **Thanks for the responses.**
> >
> > The authors give detailed responses to address my concerns, and I appreciate the additional experiments. Based on the rebuttal, I dicide to raise my score, but recommend the authors to include the responses to the paper to clarify the differences with previous works.

---

> > > ### Author Response · Authors · 2022-08-09
> > > **Re: Response to Reviewer 646o**
> > >
> > > Thank you for your revision and comment!  We have included NCC and Gated-ACML in the experiment discussions, as well as the above mentioned paper in the related works.

---

### Official Review · Reviewer_AmRN · 2022-07-11

**Rating:** 7
**Confidence:** 4
**Soundness:** 3 good
**Presentation:** 3 good
**Contribution:** 3 good

**Summary:**

This paper develops a multi-agent reinforcement learning approach with attention to enable zero-shot generalization across scales of homogeneous agents. The approach, self-organized groups (SOG), develops mechanisms for a decentralized team to form ad hoc groups. Within each group, a conductor is elected. The conductor then provides messages to each of the other members of the group. A message summary procedure is added to maximize the mutual information between personal messages and future agent trajectories. Results across StarCraft and Predator-Prey show promising results. Ablation studies are provided to show the contribution of various conductor election schemes. Zero-shot results are provided for scaling the number of agents in Starcraft.

**Questions:**

- In Lines 137-139, the paper seems to suggest that agents that communicate are not part of the CTDE paradigm; however, communication (at least communication that is range-limitted) can be to be compliant with the CTDE paradigm. For example, see Ding et al., (NeurIPS’20). Does the paper mean to suggest otherwise?
-How well does the approach perform as a function of the communication range (and associated ability of the conductor to communicate with just one or all of the agents?

Ding, Z., Huang, T. and Lu, Z., 2020. Learning individually inferred communication for multi-agent cooperation. Advances in Neural Information Processing Systems, 33, pp.22069-22079.

**Limitations:**

The paper says that limitations were addressed and to see Section 4.2. However, word “limit” and “limitation” doe not appear anywhere in the paper except in regards to limitations of prior work or the ranges of agents tested (Line 319). As such, it is hard to see where limitations are sufficiently addressed as the checklist seems to intend. Note: The paper does state some assumptions in Line 148 (“assuming that all the messages can be delivered accurately without delay and each agent can handle all the received messages at the same time”) and Line 218 (“we assume a perfect communication channel”). However, the reproducibility checklists seems to differentiate between limitation and assumptions (though, they are semantically related). It would have been nice if the paper had proactively addressed the shortcomings of the work, when it would fail, when it might be disadvantageous to use the approach, when societal harm might occur (e.g., military applications), etc.

**Strengths And Weaknesses:**

Strengths:
+ Section 3 provides a helpful background to formally define the problem being addressed.
+ Figure 3 provides a helpful illustration describing the model architecture.
+ Figure 4 shows promising results. It might be helpful to choose a color scheme that more clearly reflects that SOG and its ablations are a group, where as COPA, EMP, etc. are baselines.
+ The approach (and ablations) are evaluated in computational studies in StarCraft and Predator-Prey.
+The evaluations show positive results in answering three key questions: (1) does the approach improve coordination vs. fully-connect communication models, (2) can the approach enable zero-shot generalization, and (3) how does performance vary with the CE selection mechanism?

Weaknesses:
- The color scheme from Figure 4 to Figure 5 is inconsistent. In one, COPA is red; in the other SOG is red.
- The paper seems to have accidentally missed some prior work that have addressed related concepts of scalability and message efficiency.
1) Wang et al. (ICML) develop a MARL approach, called IMAC, that uses variational information reasoning to learn communication protocols, similar to the message summary mechanism in SOG. As such, it would have been helpful to benchmark against IMAC – particularly to compare the ability to learn efficient, information-rich messages.
2) Niu et al., (AAMAS’21) developed a MARL approach, called MAGIC, with a centralized scheduler and message processor that performs message aggregation. The Niu et al. approach is scalable (using attention), but it has the weakness that it brings some centralization as it relies on an agent to act as a message aggregator and scheduler. It would be helpful if this paper would benchmark against MAGIC, as the problem setup is analogous (or provide sufficient justification for why HetNet is not a suitable baseline).
3) Seraj et al., (AAMAS’22) developed a CTDE MARL approach (with communication) for heterogeneous agent teams, called HetNet. The approach scales (again, using attention) with the number of agents of various types and compresses messages to binary. It would be helpful if this paper would benchmark against HetNet (or provide sufficient justification for why HetNet is not a suitable baseline).

Wang, R., He, X., Yu, R., Qiu, W., An, B. and Rabinovich, Z., 2020, November. Learning efficient multi-agent communication: An information bottleneck approach. In International Conference on Machine Learning (pp. 9908-9918). PMLR.

Niu, Y., Paleja, R.R. and Gombolay, M.C., 2021, May. Multi-Agent Graph-Attention Communication and Teaming. In AAMAS (pp. 964-973).

Seraj, E., Wang, Z., Paleja, R., Martin, D., Sklar, M., Patel, A. and Gombolay, M., 2022, May. Learning efficient diverse communication for cooperative heterogeneous teaming. In Proceedings of the 21st International Conference on Autonomous Agents and Multiagent Systems (pp. 1173-1182).

Writing:
-“ which is featured with conductor election (CE) and message summary (MS)” is not grammatically quite correct. Perhaps “a message summary (MS) mechanism” would be better.
-Line 16 should be “have” instead of “has”
-Line 21, no comma before “while”
-Line 49 should have an endash before “based”
-Line 135 “like QMIX [25]” should be offset with commas
-Line 350 should have a comma after “First”
-Line 354 should not have a period after “Table”
-Line 372, it is helpful not to start a sentence with “but”
-Line 328 should have a comma after “Then”
-Line 344 – it is best not to have contractions

====== Post-rebuttal ======

I have raised my score to reflect the improvements the authors have made in their paper.

---

> ### Author Response · Authors · 2022-08-02
> **Response to Reviewer AmRN**
>
> Thank you for your constructive feedback.
>
> ### Q1: Comparison to IMAC.
> IMAC provides a regularizer to communicate under limited bandwidth. The objective is similar to our implementation of $\mathcal{L}_{FP}$, where SOG minimizes $KL(p(m)|q(m))$ and IMAC minimizes $KL(p(m)|z(m))$. The difference between $q(m)$ and $z(m)$ is that $q(m)$ is a variational estimator learned by neural network, and $z(m)$ is a fixed prior, induced by the bandwidth. We think it is unfair for IMAC to implement the IMAC objective in our environment with no bandwidth limit, since the target of IMAC is to decrease the bandwidth while that of SOG is to predict the future state. We have tried to change our $q(m)$ to a Gaussian prior $z(m)$ and find the performance degrading to qmix_atten level. Therefore we think the scheduler in IMAC is necessary for its objective. We will try to implement the scheduler and show IMAC's performance in our final version.
>
> ### Q2: Comparison to MAGIC.
> The original MAGIC uses the actor-critic structure, and can not handle the entity-wise input. But its core structure: the Scheduler and Message Processor, can be incorporated into our structure. So we implement a qmix_atten based MAGIC according to its original code, adding a one- or two-layer Scheduler and Message Processor to agent with entity-wise input. We show the results in Appendix J. We find that the 2-layer MAGIC is better than the 1-layer one. And SOG-rl shows better zero-shot generalization ability.
>
> ### Q3: Comparison to HetNet.
> HetNet aims to communicate between heterogeneous agents, and each kind of agent doesn't share parameters. However, our environment doesn't give each agent a class label, so the heterogeneity part in HetNet can hardly be compared in our setting. However, the binary message part in HetNet can be applied to our structure. Due to our limited computing resources, we run the experiments of MAGIC first. We will attach the results of binary code message in our final version.
>
> ### Q4: Writing && Color scheme inconsistent.
> We have fixed the problem you mentioned and upload a new version of paper.
>
> ### Q5: Communication with CTDE.
> Our communication condition is the same as that in Ding et al., (NeurIPS’20). As it suggests, SOG is compliant with the CTDE paradigm, since the communication is range-limited.
>
> ### Q6: Performance as a function of the communication range.
> We test the communication range on Resource Collection, sight range=0.5. Each communication range is run for 6 seeds and $8\times 10^6$ time steps. The mean+std results are as follows:
>
> |Communication Range | 0.1| 0.3|0.5|0.7|0.9
> |:--------------------------------|:-------------|:-------------|:-------------|:-------------|:-------------|
> |Test Return Mean|282.5|355.13|457.6|423.33|402.83|
> |Test Return Std|91.76|102.15|98.02|93.76|124.82|
>
> We can find that when the communication range is equal to the sight range, the agents perform best. A possible reason is that Resource Collection is a task that agents need not care about the entity that is far from itself. Therefore, messages from the far agent is not useful for the current agent and will complicate the learning, causing the performance drop.
> ### Q7: Limitations.
> We are sorry for not presenting limitations directly for saving space. As stated in the section 4.2.2, we only test SOG on the perfect communication channel. Its performance may fall down when faced with broken communication channel. Another limitation for random CE is that the expectation of the size of each group is decided by a pre-defined hyperparameter, i.e., the probability of agent elected as a conductor. When transferred to unseen scenarios, if the size of each group differs a lot from the training condition, it may cause the performance drop. We try to git rid of the hyperparameter by introducing DPP-based CE and RL-based CE, but they both require a centralized conductor elector, which is not full CTDE.

---

> > ### Comment · Reviewer_AmRN · 2022-08-06
> > **Response**
> >
> > Thanks to the authors for their rebuttal. The provided information is helpful.
> >
> > It would be helpful if the main paper included the limitations section, as asked for by NeurIPS.
> >
> > The commentary re: MAGIC and HetNet is reasonable and should be included in the related works (or results) section of the main paper.

---

> > > ### Author Response · Authors · 2022-08-09
> > > **Re: Response to AmRN**
> > >
> > > Thank you for your comment. We have included MAGIC and HetNet in the related works, and given a guidance of their experiment results and SOG's limitation discussions to the Appendix in the main paper.

---

> > > > ### Comment · Reviewer_AmRN · 2022-08-10
> > > > **Thanks**
> > > >
> > > > <eom>

---

### Official Review · Reviewer_xiGw · 2022-07-11

**Rating:** 7
**Confidence:** 4
**Soundness:** 3 good
**Presentation:** 3 good
**Contribution:** 3 good

**Summary:**

This work proposes a novel group-based communication scheme for MARL which is trained under the CTDE paradigm. Agents are periodically grouped together with each group having a single conductor which receives messages from each agent and sends back a summary of all its received messages. Messages are received through a variational encoder-decoder architecture and aim to encode information about the local trajectory of agents. Three approaches to select conductors within groups based on random selection, a policy gradient RL agent and determinal point process are proposed. The resulting Self-Organized Group (SOG) algorithm is evaluated in varying environments and demonstrates generalisation ability over varying numbers of agents and varying partial observability across training and testing tasks without further fine-tuning.

**Questions:**

1. Is the evaluated SOG algorithm based on REFIL as the MARL foundation with the additional communication paradigm? This does not quite come across clearly in Section 4.3.
2. Do all the evaluated baselines also apply parameter sharing across all agents? I would expect that parameter sharing helps significantly, in particular when communication is applied as agents follow the same communication and therefore already have a “common language” without further coordination. If not, all baselines should be (re-)evaluated with parameter sharing.
3. In Figure 4a, b, it appears that training performance in the Resource Collection task with larger sight range (1.0) is worse than in the task with shorter sight range (0.5)? This appears strange given that inputs are of the same size and in the shorter sight range case more values would simply be masked to 0s. Why do you think this appears?
4. What does shading in all training learning curves represent (standard deviation, variance, confidence intervals, …)? Same for tables 1 and 2.
5. In line 321 regarding the SMAC experiments, it is stated that a “imaginary objective” is applied as in prior work - what does this objective contain and incentivize?
6. In lines 167ff you state - “For those who receive no group invitation or leave the sight of the constructor during in-group communication, they form a group of themselves.” - it is unclear to me whether this means each of these agents is in a group alone or all these agents are together in a group? Latter would allow agents to communicate beyond boundaries of visibility.

**Limitations:**

The authors have sufficiently discussed limitations of their work.

**Strengths And Weaknesses:**

# Strengths
1. Generalisation and communication for cooperative MARL are very relevant and difficult problems.
2. The idea to group agents and efficiently communicate among agents based on agent proximity to improve coordination is intuitive and well motivated. The method is clearly outlined and well depicted as three intertwined components.
3. Generalisation to tasks with a different number of agents and varying degrees of partial observability are considered in the conducted evaluation.
4. Experiments are conducted in three different environments and comparison is made to several MARL algorithms with similar architecture and communication. Message learning components are ablated in Table 1 and further experiments investigating generalisation to varying degrees of partial observability at constant number of agents are presented in the appendix.

# Weaknesses
**Major:**
1. I have concerns regarding the novelty and significance of this work. The core components of SOG appear to be (1) identifying groups of agents to communicate through conductor election, (2) encoding messages through a variational encoder optimized with a mutual information objective, (3) summarising/ aggregating messages, and (4) being invariant to number of agents due to architecture design. Out of these, it appears to me that only (1) and (3) are novel with (4) following from the proposed architecture of REFIL [12], and (2) being proposed similarly for NDQ [A]. Also, it is worth noting that prior work already considered the problem of identifying which agents should communicate with each other for most effective communication [A, B, C, D]. The aggregation of messages in (3) is simply a sum and the conductor election process (1) was shown to be comparable to just randomly selecting an agent as conductor in experiments of this work.

2. In l. 325ff the authors state that the RL-based selection of conductors for communication improves generalisation performance compared to the random selection of conductors based on their experimental results, but the results are not conclusive and in some cases even indicate better performance for the random selection process.

**Minor:**

3. For results in the predator prey environment, it is stated that “SOG performs better when transferred to complicated evaluation scenarios” (l. 364) but generalisation returns are very similar to baselines with large variance indicating no significant difference across algorithms.
4. The downsides to CTDE described is mostly a downside of a lack of information (no specific failure/ downside of CTDE). Also the described failure case should be investigated with a simple small experiment of the described task to verify the statements, QMIX and similar algorithms are not designed for these scenarios but agents might implicitly follow a pattern to still effectively coordinate.

[A] Wang, Tonghan, Jianhao Wang, Chongyi Zheng, and Chongjie Zhang. "Learning nearly decomposable value functions via communication minimization." International conference on learning representations, 2020.

[B] Du, Yali, Bo Liu, Vincent Moens, Ziqi Liu, Zhicheng Ren, Jun Wang, Xu Chen, and Haifeng Zhang. "Learning correlated communication topology in multi-agent reinforcement learning." In Proceedings of the 20th International Conference on Autonomous Agents and MultiAgent Systems, pp. 456-464. 2021.

[C] Jiang, Jiechuan, and Zongqing Lu. "Learning attentional communication for multi-agent cooperation." Advances in neural information processing systems, 2018.

---

> ### Author Response · Authors · 2022-08-02
> **Response to Reviewer xiGw**
>
> Thank you for your constructive feedback. We will first claim your concerns about our novelty. Then we will answer the questions.
>
> ### Q1: Random CE and RL-based CE.
>
> We are so sorry that we gave the wrong legend in Fig.5(b) by mistake. Actually it's color of each algorithm is in agreement with that in Fig.5(a), therefore the legend word of "SOG" and "SOG_rl" should be exchanged in Fig.5(b). You can also validate this by comparing the performance in Fig.5(a)"3-8sz_symmetric_D" and Fig.5(b)Dispose SR=3(Training).
>
> After fixing the legend bug in the revision version, we can find that SOG_rl performs better than SOG on most dynamic teams and varying partial observability, except the varying team condition on SC2 map "3-8MMM_symmetric". A possible reason is that the "MMM" map in SC2 includes more kinds of agents, which needs more exploration on the group formation. And the random CE can be regarded as a structural exploration method. We gave more discussion of random CE in Appendix G.2.
>
> ### Q2: Novelty of our paper.
>
> (1) Effective communication is just a byproduct of our proposed SOG, which mainly focuses on the zero-shot generalization ability. Previous methods usually aim for reducing the communication cost, hardly generalize to unseen scenarios, since the change of the neighbors' number may break the learned communication patterns. On the contrary, our proposed conductor-follower mechanism can maintain a similar communication pattern to the training condition. Taking a 2-agent- to 4-agent-system generalization task as an instance, the trained 2-agent team may perform worse in a 4-agent scenario, since the agent is only trained to cooperate with another one agent, and it may be confused by the message sent by the other 2 agents. However, our proposed SOG mechanism has a high probability to divide the 4-agent into 2 groups, and then preventing the message delivery between the two groups. By doing so, in unseen scenarios, the agent may find the communication pattern is similar to that in training, and perform a similar 2-agent coordination pattern. Therefore, an organized group under the unified command of a conductor can better adapt to an unseen scenario than individuals.
>
> (2) As far as we know, we are the first to introduce a reinforcement learning-based conductor election method for multi-agent systems. As shown in the experiment and Q1, RL-based conductor election shows competitive performance.
>
> (3) Our MI-objective is totally different with that in NDQ. We introduce an MI-objective with the function of future prediction into the communication message. NDQ maximizes $I(m_i;a|\tau,m_{-i})$. The message in NDQ is expected to affect the action selection, while our objective only cares about the state and future observation trajectories, having no business with the action. NDQ also minimize $I(m;\tau)$ as an entropy regularizer, while one of our object is maximizing $I(m;\tau)$, completely opposite.
>
>
>
> ### Q3: Large variance in predator prey.
>
> Original table 1 shows the result of 6 seeds. To give a more convincing results, we re-run each ablation experiment for another 6 seeds. We update the result in the revision version by the mean+std of the medium 6 of the 12 seeds, i.e., 4th-9th seed.
>
> ### Q4: Demo depicting downsides of CTDE.
>
> We are designing a demo on gridworld environment to depict the downside of CTDE. We will attach the result in the future.
>
> ### Q5: Is SOG REFIL-based? && What is imaginary objective?
> Imaginary objective is the objective that REFIL used. It is useful in hard environment. We use REFIL as baseline in SC2, and use QMIX in another two environments.
>
> ### Q6: Parameter sharing.
> All baselines and SOG share parameters across all agents.
>
> ### Q7: In Resource Collection, larger sight range brings worse performance.
> Yes, this is a common, but strange phenomenon. COPA[D] also reports this phenomenon in experiments, and the author thinks it's because the larger sight complicates the learning. We think another reason may be the defect of parameter sharing on entity-wise input: If each agent see all the entities, they will get
> the same embedding after the attention layer, and get the same local Q. This may confuse the mixer and is harmful for coordination.
>
> ### Q8: Meaning of shading.
> In all tables and curves, the shading means std. We run each experiment with 6 seeds.
>
> ### Q9: Group strategy for single agent.
> Each of these single agents is in a group alone. Agents can not communicate beyond boundaries of visibility.
>
> ### Reference
> [D] Coach-Player Multi-Agent Reinforcement Learning for Dynamic Team Composition.

---

> > ### Comment · Reviewer_xiGw · 2022-08-08
> > **Response to Rebuttal**
> >
> > I thank the authors for their rebuttal and clarifications.
> >
> > - The corrected legend in Fig 5b clarifies my misunderstanding of the benefits of the proposed conductor election process and shows benefits in particular for the RL-based election process which consistently outperforms the DPP and random election. Also, the RL-based election process appears the most robust in transferring scenarios as shown in Fig 5b. As stated in my review, I consider the conductor-election process novel, but was previously not convinced of its significance. These changes rectify that and I am now convinced by its novelty and significance.
> > - I appreciate the author's clarifications regarding SOG's novelty. In particular the difference to prior MI objectives was helpful and I apologise for my prior misunderstanding.
> > -  I also thank the authors for running additional seeds and comparisons found in the appendix as stated in other rebuttal messages.
> >
> > Further recommendation:
> > It might be helpful to explicitly state that performance plotted in Fig 4 a,b and Fig 5 a are for testing scenarios never trained in. Showing learning curves for training performance would also be interesting to see whether SOG's advantage is already visible during training or specific to its generalisation ability.
> >
> > Given the clarifications provided in all rebuttals I will raise my score of my review. I believe this work would be a good contribution to MARL generalisation and communication.

---

> > > ### Author Response · Authors · 2022-08-09
> > > **Re: Response to xiGw**
> > >
> > > Thank you for your revision and recommendation. We attach the training return curves of SOG, NCC, MAGIC and Gated_Qatten in Appendix K. To make the curve more clear, we do not show algorithms with relatively low returns, including QMIX_atten, EMP and REFIL. The results show that SOG has no obvious superior performance than NCC or MAGIC in training, which indicates that the advantage of SOG is specific to its generalisation ability.

---

### Official Review · Reviewer_Ur4a · 2022-07-22

**Rating:** 5
**Confidence:** 4
**Soundness:** 3 good
**Presentation:** 3 good
**Contribution:** 3 good

**Summary:**

This paper approaches the problem of allow members in a team to group together by allowing “conductor agents” to temporally construct groups and then allowing messages to be passed from agents to conductors (with a variational message summariser) in a unified scheduling approach. They show that in the case of having a dynamic number of agents, and also having partial observability, this method has good zero-shot generalisation performance on standard MARL benchmark tasks like predator-prey.


**Questions:**

1. Are there any insights on the differences between the message summariser using an entropy regulariser? It intuitively seems like it should work but from the results stated it is unclear why it does not
2. Are there results on increasing the message dimension upwards of 3? Do you have thoughts on why the different message dimensions (and ability to store more bits of information) do not impact performance?


**Limitations:**

I think the conductor selection might benefit from some prior information/thought put into selecting the conductors rather than randomly selecting.

Nit: some typos throughout the paper, particularly occurrences of “aims->aim” e.g., line 196, 204


**Strengths And Weaknesses:**

1. This paper is clearly written and the motivation of this work is very sound
2. The authors spend some amount of time first outlining the downsides of centralised training and decentralised execution and why communication between agents is important and can help handle partial observability.
3. The motivation for self-organised grouping in order to build communication mechanisms that satisfy properties of being lightweight and robust is also clearly laid out.

---

> ### Author Response · Authors · 2022-08-02
> **Response to Reviewer Ur4a**
>
> Thank you for your constructive feedback.
>
> ### Q1: The effect of the entropy regularizer?
> The entropy regularizer is widely used in many RL/MARL methods, added to the RL objective[A], the agent role[B], or the communication message[C]. The purpose of our adding the entropy regularizer is to keep the diversity of the message, preventing it from falling into local optima. We regard it as a way to encourage exploration, while it may affect the accuracy of the message. In the Predator-prey environment where we take ablations, the regularizer seems taking no effect. A possible reason is that the environment is relatively easy that we need more exploitation than exploration.
>
> ### Q2: Message dimension concern.
>
> Yes, we have tested the message dimension problem in Table 2. The message dimension 10 performs a little better than the message dimension 3, while message dimension 1 has the worst performance. Dimension 10 requires a larger network and more GPU memories. Large message dimension also complicates the learning, and its performance improves a little slower during training. The most appropriate message dimension is task specific. For SC2 we need 5-dimensional message. And for Resource collection and Predator prey, messages with 3 dimensions are enough. Messages with lower dimensions may be restricted by information bottleneck and can not convey enough information.
>
> ### Q3: About random conductor election.
>
> In addition to the random CE, we also design a DPP-based CE and a PG-based CE in Section 4.2.2. In some experiments, PG-based CE performs better. For more discussion about random CE, please refer to Appendix G.2.
>
>
> ### Reference
>
>
> [A] Soft Actor-Critic: Off-Policy Maximum Entropy Deep Reinforcement Learning with a Stochastic Actor.
>
> [B] Multi-Agent Reinforcement Learning with Emergent Roles.
>
> [C] Efficient Communication in Multi-Agent Reinforcement Learning via Variance Based Control.

---

### Meta-Review · Area_Chair_xzgF · 2022-08-25

**Recommendation:** Accept
**Confidence:** Certain

**Metareview:**

The reviewers carefully analyzed this work and agreed that the topics investigated in this paper are important and relevant to the field. They generally expressed positive views on the proposed method but also pointed out a few possible limitations of this paper. One reviewer argued that the authors properly outlined the downsides of alternative approaches and the importance of communication as a way of dealing with partial observability. This reviewer, however, brought up one limitation: that the conductor selection may benefit from/require prior information. After reading the rebuttal, however, this reviewer said that the authors satisfactorily answered most of their questions, and further argued that the insights from this paper will most likely be interesting to the emergent communication community. Another reviewer claimed that this paper introduced a novel group-based communication scheme for MARL. They argued that the ideas explored here are intuitive and well-motivated. This reviewer initially believed that some of the experimental results were inconclusive (e.g., regarding the claims that RL-based selection of conductors improves performance w.r.t. to random selection). The reviewer also commented, in their original review, on the possible lack of novelty: out of four components that compose this method, two are novel, one may follow from [4], and one may follow from [A]". After carefully analyzing the authors' rebuttal, however, this reviewer increased their score: they believe that the authors' detailed rebuttal helped clarify minor concerns and that the reviewer's initially-voiced major doubts (e.g., regarding the significance of this work) were mostly rectified. Overall, this reviewer believes (post-rebuttal) that this is indeed a novel and interesting paper introducing news ideas toward communication in MARL—all of which were well executed and appropriately studied. Another reviewer expressed concerns that the paper did not discuss important prior work [1-3], but was satisfied with the authors' responses and thanked them for adding more baselines as part of the experiments. Finally, one reviewer argued that even though this is an interesting method, they still had (pre-rebuttal) three main points of concern. After reading the authors' rebuttal, this reviewer said that "the authors gave detailed responses to address my concerns and I appreciate the additional experiments". Overall, thus, it is clear that all reviewers were positively impressed with the quality of this work and look forward to an updated version of the paper that addresses the suggestions mentioned in their reviews and during the discussion phase.


**Award:**

No

---

### Decision · Program_Chairs · 2022-09-14

Accept